# The extent of nitrogen isotopic fractionation in rumen bacteria is associated with changes in rumen nitrogen metabolism

**Gonzalo Cantalapiedra-Hijar** [1]◐*, **Gonzalo Martinez-Fernandez** [2]◐*, **Evelyne Forano** [3], **Stuart E. Denman** [2], **Diego Morgavi** [1], **Christopher S. McSweeney** [2]

**1** INRAE, Université Clermont Auvergne, Vetagro Sup, UMRH, Saint-Genes-Champanelle, France, **2** Agriculture and Food, CSIRO, St Lucia, QLD, Australia, **3** INRAE, Université Clermont Auvergne, UMR 454 MEDIS, Saint-Genès-Champanelle, France

◐ These authors contributed equally to this work.
* gonzalo.cantalapiedra@inrae.fr (GCH); gonzalo.martinezfernandez@csiro.au (GMF)

**Data Availability Statement:** The sequences obtained in this paper have been deposited in the European Nucleotide Archive under the accession

## Abstract

Nitrogen use efficiency is an important index in ruminants and can be indirectly evaluated through the N isotopic discrimination between the animal and its diet ($\Delta^{15}N_{animal-diet}$). The concentration and source of N may determine both the extent of the N isotopic discrimination in bacteria and N use efficiency. We hypothesised that the uptake and release of ammonia by rumen bacteria will affect the natural $^{15}N$ enrichment of the bacterial biomass over their substrates ($\Delta^{15}N_{bacteria-substrate}$) and thereby further impacting $\Delta^{15}N_{animal-diet}$. To test this hypothesis, two independent *in vitro* experiments were conducted using two contrasting N sources (organic vs inorganic) at different levels either in pure rumen bacteria culture incubations (Experiment #1) or in mixed rumen cultures (Experiment #2). In Experiment #1, tryptone casein or ammonium chloride were tested at low (1 mM N) and high (11.5 mM N) concentrations on three rumen bacterial strains (*Fibrobacter succinogenes*, *Eubacterium limosum* and *Xylanibacter ruminicola*) incubated in triplicate in anaerobic batch monocultures during 48h. In Experiment #2 mixed rumen cultures were incubated during 120 h with peptone or ammonium chloride at five different levels of N (1.5, 3, 4.5, 6 and 12-mM). In experiment #1, $\Delta^{15}N_{bacteria-substrate}$ was lowest when the ammonia-consumer bacterium *Fibrobacter succinogenes* was grown on ammonium chloride, and highest when the proteolytic bacterial strain *Xylanibacter ruminicola* was grown on tryptone. In experiment #2, $\Delta^{15}N_{bacteria-substrate}$ was lower with inorganic (ammonium chloride) vs organic (peptone) N source. A strong negative correlation between $\Delta^{15}N_{bacteria-substrate}$ and *Rikenellaceae_RC9_gut_group*, a potential fibrolytic rumen bacterium, was detected. Together, our results showed that $\Delta^{15}N_{bacteria-substrate}$ may change according to the balance between synthesis of microbial protein from ammonia versus non-ammonia N sources and confirm the key role of rumen bacteria as modulators of $\Delta^{15}N_{animal-diet}$.

number PRJEB63213 (https://www.ebi.ac.uk/).
Likewise, the data concerning the in vitro
fermentation parameters and isotopic values have
been deposited in a Zenodo repository (https://doi.
org/10.5281/zenodo.8082282) and reported in the
manuscript.

**Funding:** The travel expenses for G. Cantalapiedra-
Hijar and G. Martinez-Fernandez between France
(Experiment #1) and Australia (Experiment #2)
were funded by the INRA-CSIRO linkage program.
Additionally, the expenses for in vitro experiments
and analysis were financed by internal funds from
the research teams involved in this study. There
was no additional external funding received for this
study.

**Competing interests:** The authors declare no
conflict of interest.

## Introduction

Nitrogen exists in the environment as two stable isotopes forms: $^{14}N$ and the less abundant and heavier $^{15}N$ isotope. Ruminant proteins are naturally $^{15}N$ enriched over their diet due to enzymatic isotopic fractionation (i.e. changes in isotopic composition of a reaction product relative to the substrate) occurring during rumen N metabolism [1] and liver amino acid catabolism [2]. The resulting N isotopic discrimination, (i.e. the uptake or assimilation by living organisms of a particular isotope in preference to another isotope of the same element) between the animal and its diet ($\Delta^{15}N_{animal-diet}$) has been shown to correlate with the animal's ability to transform the feed N into animal proteins [3–5] thus representing a promising biomarker for N use efficiency (NUE) in ruminants [6]. However, the potential of this biomarker appears to be diet-dependent [6, 7] and influenced to some extent by the rumen microbial activity [1, 7]. Yet, the precise mechanism underlying the role of the rumen microbiota on $\Delta^{15}N_{animal-diet}$ remains unclear.

Pathways that govern ammonia-N uptake [8] and release [9] from bacterial cells are thought to be mainly responsible for the N isotopic fractionation in bacteria. Both reactions favour the use of compounds containing the lighter ($^{14}N$) over the heavier isotope ($^{15}N$), the final isotopic signature of bacteria likely depends on the dominating process. Less is known about N isotope use in transamination reactions involved in the catabolism as well as synthesis of amino acids (AA) when bacteria take up and assimilate peptides for growth although the $^{14}NH_2$ appears to incorporate faster than the $^{15}NH_2$ [10]. Evidence also exists that the concentration of N (organic and inorganic) may determine the extent of the N isotopic discrimination in bacteria [1, 8]. Available information is scarce but, based on existing reports [1, 9, 11], it can be hypothesized that when rumen bacteria take up ammonia-N to synthesize their protein (improving the animal NUE) they will lower their $^{15}N$ abundance (contributing to decrease the $\Delta^{15}N_{animal-diet}$). Conversely, when rumen bacteria catabolize AA for obtaining energy the resulting ammonia release (decreasing animal NUE) is associated to a greater $^{15}N$ abundance of bacteria (contributing to increase $\Delta^{15}N_{animal-diet}$). Because rumen bacteria differ in their N metabolism it is expected their natural $^{15}N$ enrichment would reflect their prevailing ability to assimilate or release ammonia. Thus, the objective of this study was to evaluate the N isotopic fractionation occurring first in three different rumen bacteria: *Fibrobacter succinogenes*, *Eubacterium limosum* and *Xylanibacter ruminicola* (previously named *Prevotella ruminicola*). These bacteria, selected for their distinct N metabolism, were grown in pure cultures with N substrates that varied in nature and level; and second with mixed rumen bacteria grown on cellulose and the same nitrogenous substrates but at different N levels.

## Materials and methods

Two *in vitro* experiments were conducted in batch cultures (Experiment #1 at INRAE [Clermont-Ferrand-Theix; France] and Experiment #2 at CSIRO [Brisbane; Australia]) to assess the effects of rumen bacterial populations, nitrogen sources and levels on nitrogen isotopic fractionation between bacteria and substrates ($\Delta^{15}N_{bacteria-substrate}$).

Animals used as donors of rumen fluid for media preparation and as source of mixed rumen cultures were managed following animal ethics guidelines. In France, all procedures were carried out in accordance with guidelines for animal research of the French Ministry of Agriculture and guidelines for animal experimentation in the European Union (European Commission, 2010). These procedures were approved by the regional ethics committee (Auvergne-Rhône-Alpes, France) and subsequently validated by the French Ministry of Agriculture under the authorization number APAFIS#7138–2016092709177605 v6. In Australia, procedures complied with the Australian Code for the Care and Use of Animals for Scientific

Purposes (eighth edition, 2013) and was approved by the CSIRO Wildlife and Large Animal Ethics Committee, application number 2016–12. Both experiments complied with the ARRIVE guidelines [12]. Low-stress animal handling techniques were consistently employed to ensure the welfare of the animals during the collection of rumen fluid. The staff responsible for collecting rumen fluid, either through stomach tubing or rumen cannula, were highly trained and experienced in minimizing any potential adverse effects. The time taken to collect rumen fluid from each animal was always kept below 5 minutes to minimize any potential suffering. Importantly, it is worth noting that no animal sacrifice, anaesthesia or analgesia was required for these procedures.

## Experiment #1: Pure culture incubations

Three different rumen bacterial strains in terms of N metabolism were chosen for this study: i) *Fibrobacter succinogenes* S85, is a fibrolytic rumen bacterium that uses ammonia as main nitrogen source ii) *Eubacterium limosum* (DSM 20543) possesses high deamination activity and iii) *Xylanibacter ruminicola* [13] (previously named *Prevotella ruminicola*) (ATCC 19189™); *Prevotella* is a predominant rumen genus that has proteolytic activity and uses either ammonia or peptides as a N source (ammonia-N user or producer). The identity of strains and purity of cultures was verified by 16S rRNA sequencing. The effect of two nitrogen sources supplied at two levels on $\Delta^{15}N_{bacteria-substrate}$ was tested on these three strains grown in monoculture. Organic (tryptone casein) and inorganic (ammonium chloride, Sigma-Aldrich) N sources were supplied at low (1 mM N) or high (11.5 mM N) concentrations added to a culture medium (adapted from previously developed medium [14], S1 Table), which contained 2 mM of ammonia-N as the baseline level and cellobiose and starch as main carbohydrate sources. To ensure consistency between experiments involving pure strains (Exp #1) and mixed cultures (Exp #2) the culture medium was the identical between the two experiments. The strains were incubated in triplicate anaerobic batch cultures at 39˚C for 48 h (four replicates per combination of strain × source × N level plus three blanks).

Cultures were started by inoculating 50 ml of medium (S1 Table) prepared using standard anaerobic techniques [15] in 125 ml serum bottles with 1 mL of 24 h-cultures of the respective rumen bacterial strain. After 48 h of incubation, the fermentation was stopped by placing the bottles in ice. The content was centrifuged at 4˚C, 10 min, 4000 × g. Samples of supernatant were stored at − 20˚C until analysis for volatile fatty acid (VFA—0.8 mL in 0.5 mL of a 0.5 M HCl solution containing 2% [wt/vol] metaphosphoric acid and 0.4% [wt/vol] crotonic acid) and ammonia content (1 mL in 0.1 mL of 5% [wt/vol] orthophosphoric acid). The bacterial pellet and 2 mL of the media sampled before the incubation were stored at -80˚C until N isotopic analyses.

## Experiment #2: Mixed culture incubations

A second *in vitro* experiment was conducted in mixed rumen cultures to assess the effects of two N sources supplied at five levels on $\Delta^{15}N_{bacteria-substrate}$. Organic (peptone, Oxoid Ltd.) and inorganic (ammonium chloride, Sigma-Aldrich) N were used as N sources. The five levels of N used were: 1.5, 3, 4.5, 6 and 12-mM N (final concentration in the media).

A selective medium for cellulolytic bacteria with cotton thread and acid swollen filter paper as cellulose sources was used in the experiment (adapted from previously developed medium [14], S1 Table). The basal medium was adjusted to have a maximum ammonia-N concentration of 0.5 mM. The inoculum source was collected from three pasture fed steers grazing low quality pasture and stored at −80˚C in a glycerol-based cryoprotectant modified from the storage solution outlined by our colleagues [14]. Once thawed, 1.5 mL of inoculum (0.5 mL from

each glycerol tube) was inoculated in 100 ml of the selective medium for cellulolytic bacteria described above. The fibrolytic enrichment cultures were grown at 39˚C and transferred twice to new media every four days before the experiment started. Two mL of the used media were sampled before the incubations and stored at -80˚C for N isotopic analysis.

A 120-h incubation was carried out with three replicates per source × N level combination plus three blanks. The selective medium for cellulolytic bacteria was prepared using standard anaerobic techniques [15], dispensed in an anaerobic chamber with an atmosphere of 95% $CO_2$ and 5% $H_2$ (COY Laboratory Products Inc., Ann Arbor, MI) as 78 ml aliquots in 250 ml Ankom bottles (Ankom$^{RF}$ Gas Production, Ankom Technology, NY, USA) and 2.5 mL of fibrolytic enrichment cultures were inoculated per bottle. The commercial wireless system (Ankom$^{RF}$ Gas Production, Ankom Technology, NY, USA) consisting of bottles equipped with pressure sensor modules and a reception base station connected to a computer was used to release pressure [16]. No gas production measures were conducted in this in vitro experiment due to a technical problem. After 120 h of incubation at 39˚C, the fermentation was stopped by placing the bottles in ice. The content was centrifuged at 4˚ C, 10 min, 4000 × g. Samples of supernatant were stored at − 20˚C until analysis for volatile fatty acid and ammonia (1 mL content in 0.25 mL of 25% [wt/vol] metaphosphoric acid). The bacterial pellets were stored at -80˚ C until N isotopic analyses.

## DNA extractions and Illumina Miseq sequencing

DNA extractions from bacterial pellets (Experiment #2) collected at the end of the incubation were performed following previously published methodology [17]. The bacterial populations were characterized by amplifying the v4 region of the 16S rRNA gene using the established primers (F515/R806) [18]. Each DNA sample was amplified using the target specific primers and a unique barcode combination [19]. Amplification products were visualized by performing gel electrophoresis. Product quantities were calculated, and an equal molar amount of each target product was pooled. The pooled target products were run in a 1.5% agarose gel and bands were visualized and excised under blue light trans-illumination. The amplicons were gel purified with a QIAquick Gel Extraction Kit (Qiagen, Hilden, Germany) prior to submission for 2 × 250 bp Illumina MiSeq sequencing.

Paired-end short-read sequence data generated on the Illumina MiSeq was processed using the VSEARCH package [20]. De-multiplexed paired-end sequences were first passed through cutadapt for primer removal [21] and then merged prior to sequence quality filtering, followed by error correction [22], chimera checking [23], and clustering of sequences to Amplicon sequence variants (ASVs) [24]. Taxonomic classification of bacterial ASVs was done using the IDTAXA algorithm implemented in the DECIPHER R package against the SILVA SSU r132 training set [25]. Regression analysis of ASVs and their relationship to $\delta^{15}N$ was performed using the mixOmics R package [26] using Sparse Partial Least Squares (sPLS) [27]. In the initial analysis we used a stepwise process to first build a basic sPLS model that was further optimised and tuned. The tuning involves selecting the number of components that best explains the variance in the data and the number of features (ASVs) that maximise the correlation between the predicted and actual components. The number of features was limited to a maximum of 100 ASVs per component. Tuning involved cross-validation in which the data was randomly split into 10 subgroups to train sub models to predict the accuracy of these models. Selected ASV's were transformed using Centered log-ratio (CLR) and visualised using ggplot2 [28]. The sequences obtained in this paper have been deposited in the European Nucleotide Archive (https://www.ebi.ac.uk/) under the accession number PRJEB63213.

### Fermentation analysis

Concentrations of volatile fatty acids (VFAs) (acetate, propionate, n-butyrate, iso-butyrate, iso-valerate and n-valerate) were measured by gas chromatography using standardized procedures adapted to each laboratory. In experiment #1, a Perkin-Elmer Clarus 580 GC (Perkin Elmer, Courtaboeuf, France) was used with crotonic acid as internal standard [29]. In Experiment #2 the method previously published [30] was used. The ammonia-N rumen concentration were determined by using the previously published colorimetric method [31]. Total VFA and ammonia concentrations and individual VFA proportions were determined by subtracting the concentration measured in the blank samples from those measure at the end of the incubation period. In experiment #1, when corrections using blanks yielded negative values, we assigned them a value of zero.

### Natural abundance analysis of nitrogen stable isotopes

The bacterial pellets and media were freeze-dried in Eppendorf tubes. The N stable isotopic composition ($\delta^{15}$N, i.e. natural relative abundance of the rare stable isotope of N) of bacterial pellets, media and substrates (tryptone casein, ammonium chloride) was determined using an isotope-ratio mass spectrometer (Isoprime, VG Instruments, Manchester, UK) coupled to an elemental analyser (EA Isoprime, VG Instruments, Manchester, UK) or an elemental analyser (Carlo Erba NA1500) coupled to a Delta V plus isotope-ratio mass spectrometer via a Conflo V (Thermo Scientific) for experiment #1 and #2 respectively. The analyses and calculations were performed based on previously published methodology [2, 32] for experiment #1 and #2 respectively. In experiment #1, internal in-house standards (glutamic acid) were included every 5 samples of each run to correct for possible variations in the raw values determined by the mass spectrometer. Typical replicate measurement errors for these reference materials were 0.10‰. In vitro data on natural abundance of $^{15}$N and fermentation products are publicly available at DOI: 10.5281/zenodo.8082282.

### Statistical analyses

Data from experiment #1 and #2 were analysed as a univariate analysis of variance using the GLM procedure of SPSS (IBM, version 21.0), with the bottle as the experimental unit. In experiment #1, the bacterial strain, source of N, level of N and their interactions were considered as fixed effects for the analyses. In experiment #2, the level of N, source of N and their interaction were considered as fixed effects. Linear (L), quadratic (Q) and cubic (C) components of the response to incremental levels of nitrogen were evaluated using polynomial contrasts. Effects were considered significant at $P \leq 0.05$. When significant differences were detected, differences among means were tested by pairwise comparisons (LSD test).

## Results

### Nitrogen isotopic fractionation in pure culture (Experiment #1)

Both the N level (high vs low) and source (ammonium chloride vs protein) significantly influenced the $\Delta^{15}$N$_{bacteria-substrate}$ values (**Fig 1**) and ammonia-N but in a different way depending on the specific bacterial N metabolism (Table 1; bacteria × N source × N level interaction; $P < 0.001$). When the N level shifted from low to high the $\Delta^{15}$N$_{bacteria-substrate}$ increased ($P<0.001$), except when ammonium chloride was used as nitrogen source for *F. succinogenes* (Bacteria × N level interaction effect; $P < 0.001$). When the ammonia-consumer bacterium *F. succinogenes* was grown on ammonium chloride, the $\Delta^{15}$N$_{bacteria-substrate}$ observed was the lowest compared to the other strains and levels ($P<0.001$).

**Table 1. Nitrogen isotopic fractionation and fermentation products from rumen bacteria cultured with different N sources (tryptone casein [organic] vs ammonium chloride [inorganic]) and N levels (low [1.5 mM] vs high [11.5 mM]) in Experiment #1.**

| | Eubacterium limosum | | | | Fibrobacter succinogenes | | | | Xylanibacter ruminicola[1] | | | | SEM | P-values[2] | | | | | |
| | Tryptone[3] | | Ammonium chloride | | Tryptone[3] | | Ammonium chloride | | Tryptone[3] | | Ammonium chloride | | | B | S | L | B × S | B × L | B × S × L |
| | Low | High | Low | High | Low | High | Low | High | Low | High | Low | High | | | | | | | |
| $\Delta^{15}N_{\text{bacteria-substrate}}{}^{[4]}$, ‰ | -5.08^c | -3.33^b | -2.92^b | 0.87^a | -4.83^b | -2.71^a | -5.92^c | -6.75^d | -2.83^c | -0.78^b | -3.38^d | -1.62^b | 0.06 | <0.001 | 0.092 | <0.001 | <0.001 | <0.001 | <0.001 |
| Ammonia-N[5], mM | 1.44^b | 2.60^a | 0.87^c | 0.06^d | 0.62^a | 0.53^a | -0.56^b | -1.19^b | -0.18^b | 2.15^a | -1.28^c | -2.29^d | 0.06 | <0.001 | <0.001 | 0.181 | <0.001 | 0.005 | <0.001 |
| VFA[6], mM | 3.55 | 2.81 | 2.92 | 2.68 | 2.91 | 2.27 | 3.92 | 2.73 | 10.2 | 7.88 | 8.98 | 7.16 | 0.17 | <0.001 | 0.53 | 0.002 | 0.13 | 0.162 | 0.76 |
| VFA[7] mol/100 mol | | | | | | | | | | | | | | | | | | | |
| Acetate | 49.3 | 28.5 | 60.9 | 65.7 | 92.9 | 96.3 | 93.8 | 94.3 | 66.7 | 47.5 | 69.6 | 65.5 | 1.56 | <0.001 | <0.001 | 0.070 | 0.012 | 0.206 | 0.18 |
| Propionate | 4.03 | 0.00 | 5.85 | 0.00 | 3.70 | 2.06 | 4.21 | 3.55 | 32.8 | 38.7 | 30.4 | 34.5 | 0.44 | <0.001 | 0.59 | 0.67 | 0.092 | <0.001 | 0.76 |
| Iso-butyrate | 0.53 | 0.18 | 0.99 | 0.28 | 0.19 | 0.00 | 0.02 | 0.00 | 0.32^b | 3.54^a | 0.00^b | 0.00^b | 0.06 | <0.001 | <0.001 | 0.008 | <0.001 | <0.001 | <0.001 |
| Butyrate | 41.9 | 68.1 | 28.3 | 29.3 | 0.68 | 0.00 | 0.26 | 0.20 | 0.11 | 0.00 | 0.00 | 0.00 | 1.50 | <0.001 | 0.007 | 0.15 | 0.002 | 0.115 | 0.15 |
| Valerate | 0.21 | 0.10 | 0.05 | 0.06 | 0.00 | 0.00 | 0.04 | 0.00 | 0.00 | 0.00 | 0.00 | 0.00 | 0.01 | 0.009 | 0.34 | 0.42 | 0.20 | 0.77 | 0.49 |
| Iso-valerate | 4.10 | 3.05 | 3.95 | 2.88 | 2.57^a | 1.68^b | 1.63^b | 2.00^b | 0.00^b | 10.3^a | 0.00^b | 0.00^b | 0.15 | 0.002 | <0.001 | <0.001 | <0.001 | <0.001 | <0.001 |
| Caproate | 0.00 | 0.05 | 0.00 | 0.00 | 0.00 | 0.00 | 0.00 | 0.00 | 0.00 | 0.00 | 0.00 | 0.00 | 0.00 | 0.383 | 0.32 | 0.32 | 0.38 | 0.38 | 0.38 |

a,b,c,d Mean comparisons were conducted when significant interactions were detected for B × S × L. Within a row and bacteria strain mean with different letters differ significantly (P<0.05)

[1] Previously named as *Prevotella ruminicola*

[2] B = rumen bacteria strain; S = protein source; L = Nitrogen level

[3] Pancreatic digest of casein

[4] Difference in natural $^{15}N$ abundance between the isolated rumen bacteria pellets and their media (including the tested N sources) after 48h incubation

[5] Ammonia-N concentration at the end of the incubation period minus ammonia concentration of the culture medium before bacteria inoculation (ammonia-N from the 2 levels of ammonium chloride source were also subtracted)

[6] Volatile fatty acids concentration at the end of the incubation period minus volatile fatty acid of the culture medium before bacteria inoculation (blanks)

[7] When corrections using blanks yielded negative values, we assigned them a value of zero (no production).

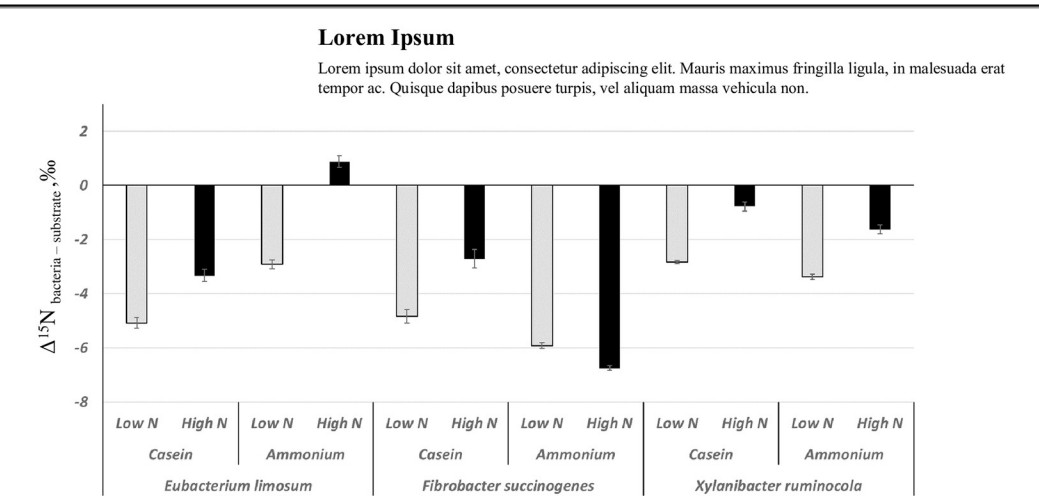

**Lorem Ipsum**

Lorem ipsum dolor sit amet, consectetur adipiscing elit. Mauris maximus fringilla ligula, in malesuada erat tempor ac. Quisque dapibus posuere turpis, vel aliquam massa vehicula non.

**Final_fig 1_202308303704.tif** This is a preview of your figure rendered on a simulated PLOS journal page.

Maecenas ac est sit amet odio sollicitudin euismod. In risus odio, convallis a neque ac, varius ultricies arcu. Vestibulum et quam iaculis, ultricies odio et, molestie magna. Suspendisse vehicula purus id turpis eleifend, et convallis dui dignissim. Praesent tempus elit a metus sollicitudin, sed fringilla nulla porttitor. Nullam in tempus massa. Nunc maximus magna massa, nec volutpat risus rhoncus ut. Fusce quis ante sem. Aenean nulla nibh, tempus sit amet rhoncus at, eleifend vel risus. Sed dictum, sem ultrices elementum pharetra, lacus diam volutpat orci, scelerisque semper dui lacus ut enim.

Suspendisse in nunc id lacus commodo consequat. Proin semper aliquam varius. Fusce vitae neque aliquam nisi ultrices sodales vitae ut enim. Vivamus nec dictum ipsum. Sed condimentum ante eu urna tincidunt tincidunt. In ac lacus nec ipsum viverra volutpat posuere vel lacus. Class aptent taciti sociosqu ad litora torquent per conubia nostra, per inceptos himenaeos. Morbi rhoncus ipsum quis lorem hendrerit, at vulputate massa tempus. Ut arcu nisl, gravida vitae risus ultricies, porta venenatis massa. Cras dignissim, enim at faucibus aliquam, sapien nisl eleifend dolor, vel mollis nulla nisi id ipsum. Pellentesque vehicula ultricies risus sit amet faucibus. Praesent sit amet mi ac est faucibus accumsan. Praesent pulvinar sit amet orci auctor feugiat.

Phasellus vitae congue est. Duis rutrum iaculis nunc, sed sollicitudin neque eleifend nec. Pellentesque ac nisi eget tortor imperdiet sagittis ut in orci. Mauris porta convallis euismod. Donec in ultricies urna, nec interdum lectus. Nullam sit amet finibus augue, eget rutrum metus. Nam faucibus, urna ac finibus eleifend, neque nisi lobortis ante, at pharetra purus purus sed urna. Curabitur sit amet dui at enim porta posuere non vehicula ligula. Suspendisse potenti. Vestibulum arcu magna, vulputate a massa ac, molestie tincidunt dui.

Donec id tempus lacus, sed tristique nulla. Nullam rutrum risus ut pharetra porttitor. Nam mattis dolor erat, sed volutpat est mattis sed. Suspendisse eu porta tellus. Cras gravida velit sed maximus fermentum. Fusce vitae metus commodo, sagittis nunc sed, faucibus nunc. Integer iaculis quam mattis, luctus neque in, viverra magna. Nulla rhoncus feugiat orci, quis posuere ligula ornare at. Integer vel sagittis risus. Donec semper metus nec finibus accumsan. Mauris sit amet suscipit ante. Aliquam accumsan, nisl vitae vulputate elementum, turpis nibh varius urna, vel bibendum nulla nunc ac quam. Aenean malesuada egestas maximus. Pellentesque faucibus, odio at tincidunt ullamcorper, eros nisi pellentesque mi, non blandit sapien neque quis lectus.

**Fig 1. Isotopic N discrimination between the pure bacteria and their substrates in in vitro cultures using different N sources (tryptone [organic] vs ammonium [inorganic]) and levels (low [1.5 mM] vs high [11.5 mM]) in trial 1.** Negative values correspond to bacteria pellets depleted in $^{15}$N relative to their substrates.

However, we did not observe higher $\Delta^{15}N_{bacteria-substrate}$ values when *E. limosum* was grown with tryptone casein as compared to ammonium chloride. Regarding *X. ruminicola*, a proteolytic bacterium, the highest $\Delta^{15}N_{bacteria-substrate}$ was observed with the high level of tryptone casein. The highest concentration of ammonia-N was observed when *E. limosum* was grown with the tryptone, followed by *X. ruminicola* and *F. succinogenes* grown on the same medium.

When ammonium-chloride was used as the sole N source, *X. ruminicola* consumed the greatest amount of ammonia-N followed by *F. succinogenes* and *E. limosum* (Table 1). Total VFAs concentration and individual profile were different (P < 0.001) between strains, with *X. ruminicola* being the highest producer of the quantified VFAs. In terms of VFA profile, acetate proportion was higher with *F. succionegens*, whereas propionate and butyrate were in greater proportion with *X. ruminocola* and *E. limosum*, respectively. The N source and level significantly (P < 0.001) affected the relative concentration of acetate, butyrate, iso-butyrate and iso-valerate. Acetate increased with the inorganic N source, whereas butyrate and branched volatile fatty acids increased with the organic source. The only significant (P < 0.001) bacteria × N source × N level interaction was observed for iso-valerate and iso-butyrate.

## Nitrogen isotopic fractionation in mixed rumen bacteria (Experiment #2)

The mixed enriched bacterial culture totally degraded the cotton cellulose source on all the treatments after 120 h of incubation. A significant (P < 0.001) interaction by N source and levels was observed for ammonia-N and VFAs (Table 2). When bacteria were cultured with inorganic N (ammonium chloride), the bacterial pellet was linearly depleted in $^{15}$N compared to the lowest level (from -2.95 to -5.50) and numerically there was a decreasing trend in $\Delta^{15}$N$_{bacteria-substrate}$ as ammonia N increased.

In contrast, when the N source was organic N (peptone) the bacterial pellet was linearly enriched in $^{15}$N compared to the lowest level of N (from -2.09 to 0.58; P > 0.05). Only the N source influenced (P < 0.001) the $\Delta^{15}$N$_{bacteria-substrate}$, with the lowest $\Delta^{15}$N$_{bacteria-substrate}$ observed with the inorganic nitrogen source. Ammonia-N concentration increased linearly

**Table 2. Nitrogen isotopic fractionation and fermentation products from batch mixed rumen bacteria cultured with different N sources (peptone [organic] vs ammonium chloride [inorganic]) and N levels in Experiment #2.**

| | Peptone | | | | | Ammonium chloride | | | | | | P-values[1] | | | |
| | 1.5 mM | 3 mM | 4.5 mM | 6 mM | 12 mM | 1.5 mM | 3 mM | 4.5 mM | 6 mM | 12 mM | SEM | L | S | L x S | Constrast[2] |
|---|---|---|---|---|---|---|---|---|---|---|---|---|---|---|---|
| $\delta^{15}$N bacterial pellet, ‰ | -2.09$^c$ | -1.38$^{bc}$ | -0.54$^b$ | -0.21$^{ab}$ | 0.58$^a$ | -2.95$^a$ | -4.03$^b$ | -4.65$^{bc}$ | -5.13$^{cd}$ | -5.50$^{cd}$ | 0.09 | 0.90 | <0.001 | <0.001 | L$^{AP}$ |
| $\Delta^{15}$N$_{bacteria-substrate}$[3], ‰ | -2.23 | -2.32 | -2.07 | -2.19 | -2.48 | -2.00 | -2.68 | -3.02 | -3.29 | -3.15 | 0.09 | 0.15 | 0.004 | 0.17 | |
| Ammonia-N[4], mM | 1.46$^e$ | 3.35$^d$ | 4.47$^c$ | 5.45$^b$ | 8.96$^a$ | 0.34$^a$ | -0.33$^b$ | -0.41$^b$ | -0.44$^b$ | -1.53$^c$ | 0.87 | <0.001 | <0.001 | <0.001 | LC$^{AP}$Q$^P$ |
| Total VFA[5], mM | 26.1$^b$ | 26.4$^b$ | 27.9 | 28.0$^b$ | 32.8$^a$ | 17.4$^c$ | 25.9$^a$ | 24.8$^a$ | 24.5$^a$ | 19.6$^b$ | 0.47 | 0.019 | <0.001 | 0.002 | L$^P$ Q$^A$ |
| VFA mol/100 mol | | | | | | | | | | | | | | | |
| Acetate | 67.1$^a$ | 67.2$^a$ | 66.5$^a$ | 65.6$^a$ | 63.3$^b$ | 65.2$^b$ | 65.1$^b$ | 69.2$^a$ | 66.9$^b$ | 65.4$^b$ | 0.23 | 0.005 | 0.38 | 0.008 | L$^P$ Q$^{AP}$ |
| Propionate | 22.4 | 21.3 | 21.2 | 20.7 | 20.8 | 24.2 | 24.1 | 20.9 | 22.9 | 23.7 | 0.17 | 0.015 | <0.001 | 0.077 | |
| Iso-butyrate | 0.90$^d$ | 0.98$^{cd}$ | 1.08$^c$ | 1.63$^b$ | 1.90$^a$ | 0.92$^a$ | 0.94$^a$ | 0.80$^b$ | 0.84$^{ab}$ | 0.91$^{ab}$ | 0.01 | <0.001 | <0.001 | <0.001 | LQC$^P$ |
| Butyrate | 6.97$^d$ | 7.40$^{cd}$ | 7.87$^{bc}$ | 8.17$^b$ | 8.78$^a$ | 7.44$^a$ | 7.17$^{abc}$ | 6.95$^c$ | 7.06$^{bc}$ | 7.57$^a$ | 0.05 | <0.001 | <0.001 | <0.001 | L$^P$ Q$^A$ |
| Iso-valerate | 1.51$^d$ | 1.81$^c$ | 1.93$^c$ | 2.22$^b$ | 2.93$^a$ | 1.56 | 1.54 | 1.34 | 1.46 | 1.39 | 0.03 | <0.001 | <0.001 | <0.001 | LQC$^P$ |
| Valerate | 1.10$^d$ | 1.25$^{cd}$ | 1.41$^{bc}$ | 1.66$^b$ | 2.23$^a$ | 1.06$^a$ | 1.15$^a$ | 0.88$^b$ | 0.88$^b$ | 0.99$^a$ | 0.03 | <0.001 | <0.001 | <0.001 | LQC$^P$ |
| Acetate:Propionate | 3.00 | 3.15 | 3.14 | 3.16 | 3.04 | 2.74 | 2.72 | 3.31 | 2.93 | 2.77 | 0.03 | 0.023 | 0.005 | 0.11 | Q |

$^{a,b,c,d}$ Within a row (for each N source) without a common letter differ, P < 0.05.

[1] L = nitrogen level; S = protein source

[2] Significant (P <0.05) linear (L), quadratic (Q) or cubic (C) effects of the response to incremental levels of nitrogen estimated by polynomial contrast for peptone $^P$ or ammonium chloride $^A$, when L x S interaction was significant.

[3] Difference in natural $^{15}$N abundance between the rumen bacteria pellets and their media (including the tested N sources)

[4] Ammonia-N concentration at the end of the incubation period minus ammonia concentration of the medium before mixed bacteria inoculation (ammonia-N from the levels of ammonium chloride source were also subtracted)

[5] Volatile fatty acids concentration at the end of the incubation period minus volatile fatty acid of the medium before mixed bacteria inoculation

(P < 0.001) from 1.46 to 9.0 mM with higher levels of peptone and decreased linearly
(P < 0.001) from 0.4 to -1.53 mM with higher levels of the inorganic N source. Total VFAs
concentration and individual profile were differed (P < 0.05) by level and nitrogen source,
with a higher concentration of total VFAs observed for the organic N source. Propionate
increased with the inorganic N source, whereas butyrate, valerate and branched VFAs
increased with the organic N source. Acetate concentration was only affected by the level of N.
All the parameters showed a significant (P < 0.001) Level x Source interaction, except for
$\Delta^{15}N$, propionate concentration and acetate to propionate ratio. The supervised multivariate
approach: sparse Partial Least Squares Discriminant Analysis (sPLS-DA), showed distinctive
correlations of selective bacterial ASVs to $\Delta^{15}N$ for each nitrogen source (Fig 2; Panels A and
B). Regression analysis of these bacterial ASVs to $\Delta^{15}N$ for each nitrogen source (Fig 2; Panels
C and D) showed the main ASVs with a strong negative correlation (coefficient of correlation
= -0.72 to -0.87) to isotopic N fractionation, suggesting a more efficient N utilization, were
classified to *Rikenellaceae_RC9_gut_group* for both nitrogen sources and *Desulfovibrio* with
the organic nitrogen ASVs. ASVs with the stronger positive correlation ($R^2$ = 0.66 to 0.82) to
$\Delta^{15}N$ were classified to the families *Veillonellaceae*, *Prevotellaceae* and unclassified bacteria
*F082* for the inorganic nitrogen source, and *Lachnospiraceae* and *Ruminococcaceaea* with the
organic nitrogen.

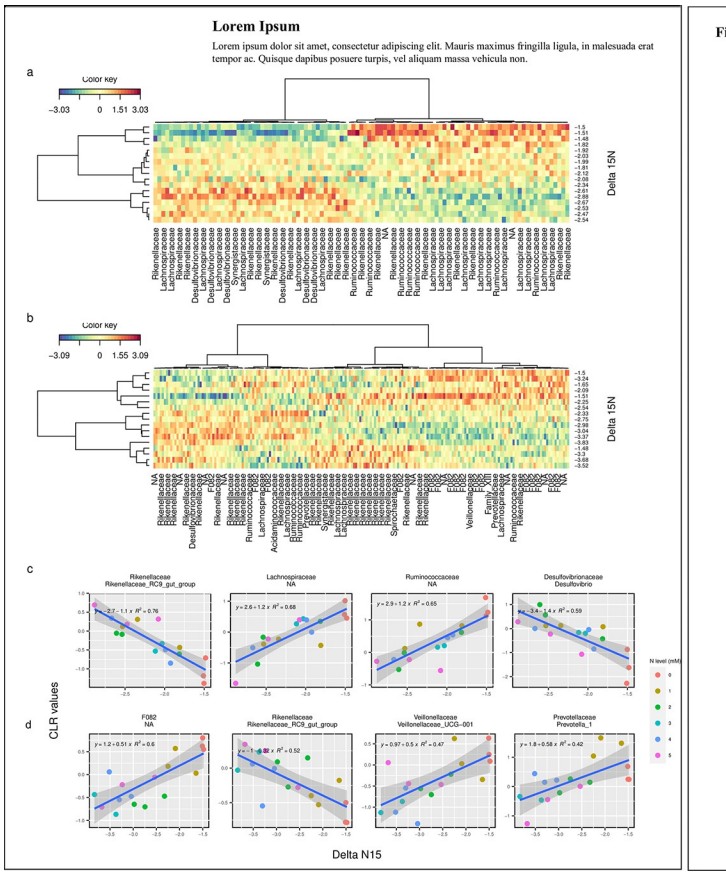

**Fig 2.** Clustering analysis using a heatmap based on the selected bacterial ASVs correlating to $\Delta^{15}N$ when peptone (A) or ammonium chloride (B) were used as
nitrogen source. Regression plots of key ASVs correlating to $\Delta^{15}N$ for peptone (C) and ammonium chloride (D). Experiment #2.

## Discussion

Rumen bacteria [1, 33] and protozoa [7] fractionate N isotopes thus contributing to the natural $^{15}$N enrichment of ruminant proteins over the diet and explains to some extent the biological basis relating $\Delta^{15}N_{animal-diet}$ to NUE in ruminants [6, 7]. Results from the present study confirm that the higher $\Delta^{15}N_{animal-diet}$ usually observed in ruminants fed high vs low N diets [34] and related to low NUE [6] would arise to some extent from the rumen bacterial activity. However, while this finding seems to be true when bacterial strains (Exp #1) were grown on organic N substrates such as tryptone, it was not always the case when ammonia was the main N source. Furthermore, although no significant interaction was observed between the N level and the N source in the batch rumen mixed culture trial (Exp#2; $P$ = 0.17), numerical trends pointed to a greater N isotopic depletion (lower $\Delta^{15}N_{bacteria-substrate}$) when the N source in mixed culture was an inorganic vs organic substrate (-1.15‰ vs -0.25‰ difference between the highest and lowest N level, respectively). Our results agree with previous studies reporting different N isotopic signatures in rumen bacteria when shifting from organic to non-organic N substrates at similar N levels [1]. We highlight a clear opposite trend in bacterial N isotopic fractionation in relation to the bacterial ability to use ammonia-N, with $^{15}$N depletions only occurring when ammonia-N is taken up by rumen bacteria. Indeed, when using a cellulolytic bacterial strain such as *Fibrobacter succinogenes* S85 (Exp #1) or alternatively a mixed cultured batch enriched for a cellulolytic bacterial population (Exp #2) greater ammonia-N levels promoted lower isotopic N discrimination. The balance between N assimilation (uptake) and dissimilation (release) is a key determinant of the N isotopic signatures of soil microbial biomass as previously demonstrated in a conceptual model [35]. Our results may further indicate that N isotopic signature in rumen bacteria may also change according to the balance between synthesis of microbial protein from ammonia versus non-ammonia N sources. In this regard, other authors [36] found that *Escherichia coli* grown on organic nitrogen sources released $NH_4^+$ and were enriched in $^{15}$N (relative to the nitrogen source) while it was highly depleted in $^{15}$N when grown on an inorganic nitrogen source. The ability of rumen bacteria to use peptides or ammonia under different nutritional contexts seems thus to determine the extent of N isotopic fractionation in the rumen.

Several metabolic pathways could be involved in the N isotopic fractionation occurring in the rumen. Although N isotopic fractionation by bacteria could theoretically originate from transport, assimilation, transfer, synthesis and excretion of N compounds [37], current knowledge suggest that the main fractionation pathways are related to bacterial ammonia assimilation and release [35, 36], the latter as a result of peptide degradation and amino acid deamination. Some of the bacterial enzymes involved in ammonia assimilation (in ruminants the glutamine synthetase [GS], glutamate synthase [GOGAT], glutamate dehydrogenase [GDH], and alanine dehydrogenase [ADH]; [38]) have been demonstrated to preferentially use $^{14}NH_4$ vs $^{15}NH_4$ [39] and explains somehow why the $^{15}$N depletion of rumen [1]; present study), soil [35] and marine [8] bacteria increases as ammonia assimilation is improved. Wang and co-workers [38] suggested that the amidation of glutamate to produce glutamine and the subsequent conversion of glutamine plus α-ketoglutarate to glutamate is the dominant pathway for ammonia assimilation in most rumen bacteria when ammonia N is low and is governed by the coupled GS-GOGAT enzymes. An alternative pathway for ammonia assimilation in some bacteria is through glutamate synthesis performed by GDH via the amination of α-ketoglutarate when ammonia-N is high [40]. However recent results indicate that both pathways for ammonia assimilation may operate concurrently within a bacterium under non-limiting ammonia-N conditions [41–43]. Less is known about N isotope use in transamination reactions involved in the catabolism as well as synthesis of amino acids when bacteria take up

and assimilate peptides for growth, although the $^{14}NH_2$ appears to incorporate faster than the $^{15}NH_2$- [10]. Transamination reactions involved in the assimilation of amino acids into protein appear to be important for the *Prevotella* genus when organic sources of nitrogen are abundant in the rumen [42]. In our study, $^{15}N$ was more depleted when *X. ruminicola* was grown on ammonia versus peptides which may indicate that the final isotopic signature of this bacterium in the rumen will reflect the net contribution of both of these N assimilation pathways. Interestingly, a recent study in dairy cows found a negative rather than positive correlation between the relative abundance of *Prevotella* taxa in the rumen and natural abundance of $^{15}N$ in plasma [44], which may suggest that in those particular conditions *Prevotella* species could be a net ammonia user rather than ammonia producer. Indeed, *Prevotella* is a very diverse genus, based on genetic diversity and also phenotypic diversity of the cultured rumen isolates [45–47]. In addition, the relative abundance of the various *Prevotella* taxa may vary according to the diet [45].

According to other study [38] there exists a correlation between the *Prevotella ruminicola* and *Fibrobacter succinogenes* populations and GDH and ADH activities, respectively. These two enzymes have been shown to be discriminant against $^{15}N$ in the bovine liver and in *Bacillus subtilis* cultures, respectively [48]. This may explain why in our study at similar N levels the $^{15}N$ natural abundances in *Prevotella ruminicola* and *Fibrobacter succinogenes* (Exp #1) were lower with ammonia chloride vs tryptone sources. Interestingly, this pattern was not observed with the third bacterium, *Eubacterium limosum*, having the greater ammonia production in our conditions even when ammonia chloride was the N source. *Eubacterium limosum* is one of the few acetogens that can produce butyrate [49] as observed in our study from their higher butyrate molar proportion compared to the other two strains. This rumen strain is very versatile in utilization of N sources, such as peptides, ammonia and single amino acids [50]. However, the greater $^{15}N$ enrichment of *E. limosum* when ammonium chloride replaced tryptone (a finding not observed with the other strains) might indicate its poor ability to use inorganic N sources.

Other colleagues [8] reviewed some studies reporting N isotope fractionation in several microorganisms using ammonia for growth, including bacteria, and found that microbial biomass was always $^{15}N$ depleted compared to ammonia substrate. The enzymes GDH and GS as well as membrane $NH_4$ transport were believed to be the main fractionation pathways involved in ammonia assimilation by marine bacteria [8, 51]. *Fibrobacter succinogenes* is one of the major fibrolytic rumen bacteria and it is known that the strain S85 of this species utilizes ammonia as its sole nitrogen source [52, 53]. However, Atasoglu [54] demonstrated that another strain of *Fibrobacter succinogenes* (strain BL2) was able to utilise non-ammonia nitrogen at high concentrations of peptides and amino acids. Furthermore, the main pathways of ammonia assimilation (GDH, ADH and GS) for *Fibrobacter succinogenes* S85 have been identified using nuclear magnetic resonance [55] and genome sequencing [56]. These findings support the $^{15}N$ depletion observed with *Fibrobacter succinogenes* S85 in the current study, and indicates that this rumen microorganism might preferentially use $^{14}NH_4$ vs $^{15}NH_4$.

The 16S rDNA amplicon sequencing analysis (Experiment #2) revealed ASVs strongly correlated to $\Delta^{15}N$ for both nitrogen sources. Several ASVs classified to the RC9_gut_group showed a negative relationship with $\Delta^{15}N$ for both nitrogen sources. A recent published study [57] indicated that *RC9* possess fibrolytic activity in the rumen and identified genes encoding saccharolytic activity against mannan, xylan and pectin, however to the best of our knowledge the nitrogen metabolism of this bacterial group has not been studied. In experiment #1 another fibrolytic bacterium, *Fibrobacter succinogenes*, promoted lower isotopic N discrimination which indicates that fibrolytic bacteria may be involved in $^{15}N$ depletion and more efficient N utilization at the rumen level. On the other hand, F082 group appeared to be associated with a

less efficient nitrogen utilization at the rumen level when inorganic nitrogen was provided. Interestingly, RC9 and F082 have been found to increase in the rumen of cattle grazing in dry tropical rangelands during the dry season [32], when nitrogen sources are scarce and inorganic nitrogen is being supplemented to animals. Further research needs to be done to determine the role of these novel rumen bacterial populations in nitrogen utilization and fractionation in animals under different feeding conditions and nitrogen sources.

## Conclusion

Our results confirmed that N isotopic fractionation by rumen bacteria may change according to the balance between synthesis of microbial protein from ammonia versus non-ammonia N sources and suggest that the extent of ammonia uptake and release strongly affects the N isotopic signature of rumen bacteria. Interestingly, a stronger $^{15}$N depletion occurred when a rumen fibrolytic bacterium specialised in ammonia assimilation was grown on non-organic N sources compared to other bacteria. These results confirm the key role of rumen bacteria as modulators of N isotopic fractionation and the subsequent link with N use efficiency in ruminants. Also, results strongly suggest that this contribution may be dependent on the type of diet used and the predominant rumen bacteria populations. Further studies should determine whether natural $^{15}$N abundances in rumen bacteria or ruminant body proteins could also reflect the efficiency of N utilization for bacterial protein synthesis in the rumen.

## Supporting information

**S1 Table. Composition of in vitro media for experiment #1 and #2.**
(DOCX)

## Acknowledgments

We gratefully acknowledge W. Smith and R. Viana Costa for their technical support in the lab and R. Andy and C. Chantelauze for analysing the natural abundance of nitrogen stable isotopes.

## Author Contributions

**Conceptualization:** Gonzalo Cantalapiedra-Hijar, Gonzalo Martinez-Fernandez, Diego Morgavi, Christopher S. McSweeney.

**Data curation:** Gonzalo Cantalapiedra-Hijar, Gonzalo Martinez-Fernandez.

**Formal analysis:** Gonzalo Cantalapiedra-Hijar, Gonzalo Martinez-Fernandez.

**Funding acquisition:** Gonzalo Cantalapiedra-Hijar, Evelyne Forano, Diego Morgavi, Christopher S. McSweeney.

**Investigation:** Gonzalo Cantalapiedra-Hijar, Gonzalo Martinez-Fernandez, Diego Morgavi, Christopher S. McSweeney.

**Methodology:** Gonzalo Cantalapiedra-Hijar, Gonzalo Martinez-Fernandez, Stuart E. Denman, Diego Morgavi, Christopher S. McSweeney.

**Project administration:** Gonzalo Cantalapiedra-Hijar, Gonzalo Martinez-Fernandez.

**Resources:** Gonzalo Cantalapiedra-Hijar, Gonzalo Martinez-Fernandez, Evelyne Forano, Christopher S. McSweeney.

**Software:** Gonzalo Martinez-Fernandez, Stuart E. Denman.

**Supervision:** Gonzalo Cantalapiedra-Hijar, Gonzalo Martinez-Fernandez, Diego Morgavi, Christopher S. McSweeney.

**Validation:** Gonzalo Cantalapiedra-Hijar, Gonzalo Martinez-Fernandez, Evelyne Forano, Stuart E. Denman, Diego Morgavi, Christopher S. McSweeney.

**Visualization:** Gonzalo Cantalapiedra-Hijar, Gonzalo Martinez-Fernandez.

**Writing – original draft:** Gonzalo Cantalapiedra-Hijar, Gonzalo Martinez-Fernandez.

**Writing – review & editing:** Gonzalo Cantalapiedra-Hijar, Gonzalo Martinez-Fernandez, Evelyne Forano, Stuart E. Denman, Diego Morgavi, Christopher S. McSweeney.

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
