## [Decision Letter · Decision Letter 0]

29 May 2023

PONE-D-23-04164The extent of nitrogen isotopic fractionation in rumen bacteria is associated with changes in rumen nitrogen metabolismPLOS ONE

Dear Dr. Cantalapiedra-Hijar,

Thank you for submitting your manuscript to PLOS ONE. After careful consideration, we feel that it has merit but does not fully meet PLOS ONE’s publication criteria as it currently stands. Therefore, we invite you to submit a revised version of the manuscript that addresses the points raised during the review process.

We look forward to receiving your revised manuscript.

Kind regards,

Adham A. Al-Sagheer

Academic Editor

PLOS ONE

Journal Requirements:

https://www.researchgate.net/figure/Daily-methane-emissions-of-wethers-n-6-before-during_fig3_234105896

https://assets.researchsquare.com/files/rs-2350552/v1/0d1d0a41-6555-439f-8a88-477693608894.pdf?c=1671033111

In your revision ensure you cite all your sources (including your own works), and quote or rephrase any duplicated text outside the methods section. Further consideration is dependent on these concerns being addressed.

This research was partially funded by the INRA-CSIRO linkage program.

The authors declare no conflict of interest. 

6. We note that you have indicated that data from this study are available upon request. PLOS only allows data to be available upon request if there are legal or ethical restrictions on sharing data publicly. For more information on unacceptable data access restrictions, please see http://journals.plos.org/plosone/s/data-availability#loc-unacceptable-data-access-restrictions. 

Reviewers' comments:

Reviewer's Responses to Questions

**Comments to the Author**

1. Is the manuscript technically sound, and do the data support the conclusions?

Reviewer #1: Yes

Reviewer #2: Yes

2. Has the statistical analysis been performed appropriately and rigorously? 

Reviewer #1: Yes

Reviewer #2: Yes

3. Have the authors made all data underlying the findings in their manuscript fully available?

Reviewer #1: No

Reviewer #2: No

4. Is the manuscript presented in an intelligible fashion and written in standard English?

Reviewer #1: Yes

Reviewer #2: Yes

5. Review Comments to the Author

Reviewer #1: The paper reports on two interesting studies, which confirm the much earlier work of Wattiaux and Reed. The studies appear well conducted and analysed,. My comments mostly relate to improving the text:

Line 19 ‘Nitrogen use’ is not an index – better ‘Nitrogen use efficiency’

Line 21 can delete ‘Evidence also exists that’

Line 23 ‘have an impact on’ = ‘affect’

Lines 41-41 section in parentheses is not needed

Line 62 is this correct (wouldn’t it increase?)

Line 102 bottles

Line 117/119 these two sentences should be merged (not sure why the sentence at lines 118-119 sits between them)

Line 204 superscripting

Table 1 & 2 Need to be clearer that ammonia and VFA values are corrected for initial values (e.g. in the Table heading

rather than just as footnotes). Were the proportions of different VFA and the acetate:propionate ratio also

corrected for initial values?

Line 262 is there any issue with multiple analysis here? Was there any need to correct for the fact that you could

explore these relationships with many hundreds of microbial taxa and so identify spurious significant

correlations?

Line 381 confirmed – as these effects were shown previously by Wattiaux and Reed

Reviewer #2: Thank you for conducting this study, it is a nice study. I have attached my comments in a separate file. Please respond them carefully. Make sure that all the data generated by the sequencing are uploaded one suitable database.

best wishes

6. PLOS authors have the option to publish the peer review history of their article (what does this mean?). If published, this will include your full peer review and any attached files.

Reviewer #1: No

Reviewer #2: **Yes: **Alaa Emara Rabee

---

## [Author Response · Author response to Decision Letter 0]

7 Jul 2023

Reviewer #1: The paper reports on two interesting studies, which confirm the much earlier work of Wattiaux and Reed. The studies appear well conducted and analysed. My comments mostly relate to improving the text:

[Authors]: Thank you for providing us with your valuable comments on our research article. We are confident that the revised version of the article now encompasses a more comprehensive and robust analysis, addressing the concerns you raised.

Line 19 ‘Nitrogen use’ is not an index – better ‘Nitrogen use efficiency’

[Authors]: Corrected

Line 21 can delete ‘Evidence also exists that’

[Authors]: Removed

Line 23 ‘have an impact on’ = ‘affect’

[Authors]: Changed to “affect”

Lines 41-41 section in parentheses is not needed

[Authors]: Considering the length of the sentence and the need to establish clear definitions for different concepts, we would prefer to retain the parentheses in this case. 

Line 62 is this correct (wouldn’t it increase?)

[Authors]: You are right, thank you for bringing this mistake to our attention. 

Line 102 bottles

[Authors]: Corrected

Line 117/119 these two sentences should be merged (not sure why the sentence at lines 118-119 sits between them)

[Authors]: Thanks for the suggestion. We have removed the second sentence and explained better in S1 Table how the medium was prepared. 

Line 204 superscripting

[Authors]: Done

Table 1 & 2 Need to be clearer that ammonia and VFA values are corrected for initial values (e.g. in the Table heading rather than just as footnotes). Were the proportions of different VFA and the acetate:propionate ratio also corrected for initial values?

[Authors]: We appreciate your suggestion. We would like to clarify that while we did correct total VFA and ammonia concentrations for their initial values, we have now realized that the individual VFA proportions in experiment #1 (only Table 1) were not calculated by subtracting the initial concentrations. We apologize for this oversight. In response to this, we have revised our manuscript. The new values of individual VFA proportions, where initial concentrations were subtracted from the final concentration, are now included in Table 1. We have provided a clearer explanation of this calculation in the Materials and Methods (section L179-182) and changed the table heading according to your suggestion. 

Line 262 is there any issue with multiple analysis here? Was there any need to correct for the fact that you could explore these relationships with many hundreds of microbial taxa and so identify spurious significant correlations?

[Authors]: The final model and analysis for correlation of ASVs to metabolite data is performed on an identified subset of ASVs. In the initial analysis we used a stepwise process to first build a basic sPLS model that is further optimised and tuned. The tuning involves selecting the number of components that best explains the variance in the data and the number of features (ASVs) that maximise the correlation between the predicted and actual components. The number of features was limited to a maximum of 100 ASVs per component. Tuning involved cross-validation in which the data was randomly split into 10 subgroups to train sub models to predict the accuracy of these models. The final parameters were then used to identified the ASVs that are presented here. This explanation appears now in the Material and Methods section (L164-171).

Line 381 confirmed – as these effects were shown previously by Wattiaux and Reed

[Authors]: Right, this has been changed. Thank you 

The extent of nitrogen isotopic fractionation in rumen bacteria is associated with changes in rumen nitrogen metabolism

Thank you for sending this manuscript to me. The manuscript discusses the effect of N sources and levels on the Δ15Nbacteria-substrate of pure and mixed bacterial rumen bacteria. The idea is brilliant and the manuscript provides more details about the N metabolism by rumen bacteria. The manuscript still needs some improvements to be published.

[Authors]: We extend our heartfelt appreciation to you for taking the time to provide us with your invaluable comments regarding our research article. The revisions made to the article have resulted in a more robust analysis, effectively addressing the concerns you raised.

Abstract:

Could you make the experimental design clearer in the abstract?

[Authors]: Considerable efforts have been made to enhance the clarity of the experimental design within the abstract (L27-L32)

Please write the full name of bacterial strains that were used.

[Authors]: Done

Introduction:

Line 60: Could you explain the AA?

[Authors]: We have implemented the use of the abbreviation "AA" for amino acids when it is first mentioned in the text

Material and methods

Why didn’t you use any component of animal diet (hay, soya meal) as substrate in the bacterial incubation besides the organic and inorganic nitrogen sources?

[Authors]: The objective of this study was to assess the nitrogen isotopic fractionation in both pure and mixed cultures across varying levels of organic and inorganic nitrogen. To maintain precise control over the nitrogen supply, we opted not to include typical animal feed such as hay or soya meal as substrates. This decision was influenced by the inherent variability and heterogeneity of these feeds, which would have posed significant challenges in maintaining consistent nitrogen levels. For the experimental setup, we utilized a modified version of a previously published medium for rumen bacteria, as described by McSweeney et al. [40]. In the case of mixed cultures, cellulose sources such as cotton thread and acid swollen filter paper were included. As for the pure cultures, cellobiose and starch were utilized as the primary carbohydrate sources within the medium (as indicated in S1 Table). These selections were made to ensure appropriate nutrient availability and create suitable conditions for the respective cultures. By employing these specific methodologies and carefully controlling the nutrient sources, we aimed to obtain accurate and reliable results regarding the nitrogen isotopic fractionation in our study.

Why the authors used a mixture of VFAs in the growth of the pure strains, the pure strains do not need all those VFAs in the growth. Also, the authors should explain the presence of all the VFAs in the supernatant of the pure culture even after the subtraction of the quantity of VFAS of the incubation medium, because the single pure culture cannot produce all the VFAs. This point should be explained in the discussion.

[Authors]: To ensure consistency between the experiments involving pure strains and mixed cultures, we utilized a mixture of VFAs (volatile fatty acids) in all the media. This approach was taken with the intention of using a similar medium for both experiments. This justification is now clearly stated in the material and methods (L103-106). The composition of the media was based on a previously published medium for rumen bacteria, which was adapted from McSweeney et al. [40]. We would like to acknowledge that the reviewer's observation regarding the pure culture's inability to produce all the VFAs is correct, and we appreciate their attention to detail in highlighting this point. Indeed, the corrections made were solely focused on the total VFA concentrations and did not account for individual proportions. We apologize for this oversight. In the revised manuscript proportions of individual VFA from both experiments are corrected by subtraction of VFA from the medium. We thank the reviewer for bringing this issue to our attention, and we apologize for any confusion caused by the initial oversight. 

Results

The authors should include a complete image of the effect of nitrogen sources on the alpha diversity and relative abundance of the bacterial community in mixed culture.

Where is the data on gas production and gas fraction besides the changes in the archaeal community in the mixed culture? This data could explain the relationship between nitrogen sources and methane and other gas production.

[Authors]: We appreciate your comment and would like to address the issue raised. Regrettably, we encountered a technical problem that prevented us from measuring gas production in the mixed culture experiment. However, we would like to emphasize that gas production is not inherently associated with N (nitrogen) isotope fractionation, and therefore, its absence in this particular experiment does not pose a significant problem. To ensure transparency and clarity, we have explicitly stated in the manuscript (lines 140-141) that gas production measurements were not conducted in the mixed culture experiment.

Discussion

The effect of nitrogen sources on the diversity and relative abundance of microbial communities should be discussed.

[Authors]: Thanks for your suggestion regarding the discussion of diversity and relative abundances of the microbial communities. While we appreciate the importance of these aspects, we have made a conscious decision not to include them in our study. This decision stems from the fact that our analysis focused on an in vitro culture endpoint trial, and therefore, caution must be exercised when extrapolating these results to in vivo conditions. The primary objective of our manuscript was to investigate the relationship with the N isotopic fractionation, and we believe that delving into the microbial diversity differences between the two sources could potentially divert the reader's attention, dilute the main storyline, and complicate the overall discussion. Consequently, we have concluded that the analyses included in the current version of the manuscript are the most relevant for addressing our research objectives.

The authors should justify the presence of mixed VFAs in the culture of single strains as the single strain produces one or more VFAs.

[Authors]: Individual VFA values were corrected in the revised manuscript (see above). Individual VFA from monocultures are now in agreement with the expected production from the strains. 

It is not necessary to repeat the sentence “previously named Prevotella ruminicola”.

[Authors]: corrected

---

## [Decision Letter · Decision Letter 1]

25 Aug 2023

The extent of nitrogen isotopic fractionation in rumen bacteria is associated with changes in rumen nitrogen metabolism

PONE-D-23-04164R1

Dear Dr. Cantalapiedra-Hijar,

We’re pleased to inform you that your manuscript has been judged scientifically suitable for publication and will be formally accepted for publication once it meets all outstanding technical requirements.

Kind regards,

Adham A. Al-Sagheer

Academic Editor

PLOS ONE

Additional Editor Comments (optional):

Reviewers' comments:

Reviewer's Responses to Questions

**Comments to the Author**

1. If the authors have adequately addressed your comments raised in a previous round of review and you feel that this manuscript is now acceptable for publication, you may indicate that here to bypass the “Comments to the Author” section, enter your conflict of interest statement in the “Confidential to Editor” section, and submit your "Accept" recommendation.

Reviewer #2: All comments have been addressed

2. Is the manuscript technically sound, and do the data support the conclusions?

Reviewer #2: Yes

3. Has the statistical analysis been performed appropriately and rigorously? 

Reviewer #2: Yes

4. Have the authors made all data underlying the findings in their manuscript fully available?

Reviewer #2: Yes

5. Is the manuscript presented in an intelligible fashion and written in standard English?

Reviewer #2: Yes

6. Review Comments to the Author

Reviewer #2: Thank you for addressing the comments.

the manuscript was improved substantially.

I think its in publishable form.

Thanks

7. PLOS authors have the option to publish the peer review history of their article (what does this mean?). If published, this will include your full peer review and any attached files.

Reviewer #2: **Yes:**

---

## [Editor Report · Acceptance letter]

31 Aug 2023

PONE-D-23-04164R1 

The extent of nitrogen isotopic fractionation in rumen bacteria is associated with changes in rumen nitrogen metabolism 

Dear Dr. Cantalapiedra-Hijar:

I'm pleased to inform you that your manuscript has been deemed suitable for publication in PLOS ONE. Congratulations! Your manuscript is now with our production department. 

Kind regards, 

on behalf of

Dr. Adham A. Al-Sagheer 

Academic Editor

PLOS ONE